# Epidemiological Analysis of Global and Regional Lung Cancer Mortality: Based on 30-Year Data Analysis of Global Burden Disease Database

**DOI:** 10.3390/healthcare11222920

**Published:** 2023-11-07

**Authors:** Xiaoxia Ji, Jingxian Chen, Junjun Ye, Shuochun Xu, Benwei Lin, Kaijian Hou

**Affiliations:** 1Medical College, Shantou University, Shantou 515031, China; jxx2439@126.com (X.J.); 19jjye@stu.edu.cn (J.Y.); 13531192521@163.com (S.X.); 2School of Public Health, Shantou University, Shantou 515063, China; 15206086545@163.com; 3School of Physiology, Pharmacology and Neuroscience, University of Bristol, Bristol BS8 1QU, UK; benlin815@gmail.com

**Keywords:** global and regional, lung cancer, mortality, epidemiology, trends

## Abstract

The objective of this study was to understand dynamic global and regional lung cancer fatality trends and provide a foundation for effective global lung cancer prevention and treatment strategies. Data from 1990 to 2019 were collected from the Global Burden Disease (GBD) database and statistical analysis was conducted using Excel 2010. Standardization was based on the GBD’s world population structure, and the Average Annual Percentage Change (AAPC) was calculated using Joinpoint 4.8.0.1 software. Bayesian age–period–cohort analysis (BAPC) predicted global lung cancer mortality from 2020 to 2030. In 2019, worldwide lung cancer deaths reached 2,042,600, a 91.75% increase from 1990 (1,065,100). The standardized age-specific death rate in 2019 was 25.18 per 100,000. Males had a rate of 37.38 while females had 14.99. Men saw a decreasing trend while women experienced an increase. High- and medium–high-SDI regions had declining rates (−0.3 and −0.8 AAPCs) whereas middle-, low-, and low–middle-SDI regions had increased mortality rates (AAPC = 0.1, AAPC = 0.37, AAPC = 0.13). Several regions, including Oceania, South Asia, East Asia, Western Sub-Saharan Africa, Southeast Asia, and Eastern Sub-Saharan Africa, witnessed rising global lung cancer mortality rates (*p* < 0.01). The global standardized mortality rate for lung cancer is expected to decrease from 2020 to 2030, but predictions indicate increasing female mortality and decreasing male mortality. Despite overall declines, rising female mortality remains a concern. Effective measures are essential to reduce mortality rates and improve patients’ quality of life in the global fight against lung cancer.

## 1. Introduction

Lung cancer is a malignant tumor, mainly referring to malignant tumors that originate in the trachea, bronchi, and lungs [1].

Currently, lung cancer poses a significant threat to human life and health. The mortality associated with lung cancer is a significant bottleneck in the extension of human lifespan [2,3]. As per the worldwide cancer data [4], by the year 2020, lung cancer constituted 2.2 million fresh cases each year, amounting to 11.7% of all global cancer incidents. It also accounted for nearly one fifth of all deaths from cancers. Pulmonary cancer has had the greatest global mortality rate among other malignant tumors. Consequently, lung cancer has been a disease that has attracted much attention for a long time. Such statistics emphasize the urgent need to address this issue comprehensively.

Understanding the determinants of lung cancer incidence and mortality is a complex challenge. The incidence of lung cancer is influenced by multiple factors, and there are significant disparities in lung cancer mortality rates among diverse regions and population groups. The geographic and social distribution of lung cancer, including epidemiological characteristics in different regions, such as age, gender, tobacco use, air pollution, and economic status, have an impact on its mortality rates [5]. According to recent medical research reports [6,7,8,9], if the nature and symptoms of cancer are correctly identified in the early stages, it can be curable [10]. Certain factors, such as smoking, exposure to air pollution, and settling in contaminated areas, can negatively influence the neoplastic prognosis for patients and their quality of life [11].

In this research, we mainly adopted two methodological approaches to explore the spatio-temporal trends in the incidence and mortality rates of lung cancer. Joinpoint Regression Analysis was used to scrutinize the temporal contours of lung cancer incidence and mortality rates from 1990 to 2019. This meticulous analysis yielded the calculation of the Annual Percentage Change (APC) and the Average Annual Percentage Change (AAPC) along with their respective 95% confidence intervals, thus affording the means to discern prevailing trajectories in lung cancer rates [12,13]. Additionally, Bayesian age–period–cohort analysis (BAPC) was conducted to forecast the future lung cancer mortality rates on a global scale. By harmonizing data extracted from the Global Burden of Disease (GBD) repository encompassing the time span from 1990 to 2019, and leveraging population projections, our study proffered a vista into lung cancer mortality forecasts extending up to the year 2030. The BAPC model, fortified with the integrated nested Laplace approximation (INLA) algorithm, facilitated the estimation of cryptic parameters and the inference of pivotal insights into the complex interplay of age, time period, and generational factors on lung cancer incidence [14,15,16]. This methodology endowed us not only with the retrospective capacity to scrutinize historical data but also with the visionary prowess to predict forthcoming trends, an asset of inestimable value for the edifice of public health stratagems. It is essential to understand the epidemiological trends and factors contributing to lung cancer mortality to develop effective strategies for prevention and treatment. In this study, we aimed to delve deeper into these trends, providing valuable insights for both the scientific community and the general public.

In summation, our empirical exploration furnished a comprehensive exegesis of the variegated tapestry of lung cancer mortality trends, underpinned by the bedrock of Global Burden of Disease (GBD) data and underpinned by the scaffolding of robust statistical methodologies. The tandem deployment of Joinpoint Regression Analysis and Bayesian age–period–cohort analysis has yielded an exhaustive elucidation of the determinants underpinning lung cancer mortality, both within the annals of history and in the foreseeable future. This scholarly endeavor aspires to underpin the formulation of efficacious prophylactic and therapeutic modalities while concurrently elevating public consciousness concerning the exigent quandaries surrounding lung cancer. By augmenting our comprehension of the epidemiology and mortality trends germane to lung cancer, we take momentous strides towards the amelioration of this formidable public health conundrum.

## 2. Materials and Methods

### 2.1. Data Sources

The data come from the 2019 GBD database, which contains all GBD diseases, risks, etiologies, injuries, natural injuries, and sequelae syndromes. The objective of GBD studies is to furnish comprehensive and equivalent evaluations of global health, considering factors like causes of death, disability, and associated risk elements. The GBD Study of 2019 evaluated death rates, incidence, prevalence, and DALYs for an aggregate of 369 causes of illness and impairment, along with 87 risk factors or clusters. This was achieved at various scales: globally, regionally, and across 204 geographically similar and proximate countries and regions, which were further divided into 21 regions and 5 groups based on the Socio-Demographic index (SDI)—high-SDI, medium–high-SDI, medium-SDI, medium–low-SDI, and low-SDI regions. The spatial data infrastructures of countries were estimated based on economic growth, fertility, and educational attainment. The 2019 edition of GBD is the latest edition, superseding all previous editions [17]. Individualized data can be obtained by using the GHDx inquiry tool (https://vizhub.healthdata.org/gbd-results/, accessed on 30 October 2023). This research involved the selection of data from the global, high-SDI, medium–high-SDI, medium-SDI, medium–low-SDI, and low-SDI regions, along with data from 21 regions as defined by the GBD’s regional categorization system, for analysis.

### 2.2. Joinpoint Regression Analysis

This study utilized the Joinpoint 4.8.0.1 software available on the website of the National Cancer Institute (https://surveillance.cancer.gov/joinpoint/download, accessed on 30 October 2023) to perform segmented regression analysis using logarithmic linear regression [12]. The focus of the examination was on the patterns in lung cancer incidence and mortality rates from 1990 to 2019. In order to represent the course and extent of the trends, the study computed the APC and the AAPC. The APC and AAPC figures, along with their 95% confidence intervals (CI) [13], if found to be greater than 0, signified an uptrend within the respective periods; on the other hand, values falling below 0 demonstrated a downtrend.

### 2.3. Bayesian Age-Period-Cohort Analysis

The Bayesian age–period–cohort analysis (BAPC) technique was utilized to project the global mortality rate of lung cancer from 2020 through 2030. BAPC considers the effects of age, time period, and generational cohorts on disease outcomes such as lung cancer mortality.

This forecast relied on data concerning worldwide lung cancer mortality from the Global Burden of Disease (GBD) database, covering the time span from 1990 to 2019 [14]. The future standardized population was estimated using population projections from the 2017 GBD database [15].

The BAPC model is a sophisticated statistical tool that combines prior information about unknown parameters with sample information, resulting in the estimation of posterior distributions and the inference of these unknown parameters. To approximate the posterior marginal distribution efficiently, we adopted the integrated nested Laplace approximation (INLA) algorithm.

Given that the impacts of sequential time frames might be comparable, we utilized a second-order random walk (RW2) model. This allowed us to delve into the influence of various factors, including age brackets, time periods, and generational groups, on the number of incidences, specific age incidence rates, and normalized incidence rates [16]. For the visual representation and interpretation of the results obtained through the BAPC analysis, we employed the R4.2.1 software. This widely recognized statistical software facilitated the generation of visual outputs that aid in understanding the complex relationships and trends identified in the data.

## 3. Results

### 3.1. Epidemiology of Global Lung Cancer Mortality

#### 3.1.1. Overall Overview of Lung Cancer Mortality Worldwide in 2019

According to the analysis of the database results, in 2019, the global number of deaths from lung cancer was 2.0426 million, representing a 91.75% increase compared to the 1990 death toll of 1.0651 million. However, to better understand these trends, it is essential to consider the impact of changes in population demographics, specifically increased life expectancy and shifts in age distribution. This is particularly relevant in the interpretation of lung cancer mortality statistics. In 2019, lung cancer resulted in 1.3861 million male fatalities and 0.6565 million female fatalities. These represent a 75.28% and 139.37% surge, respectively, from the 1990 statistics of 0.7907 million male deaths and 0.2743 million female deaths (refer to Table 1). It is crucial to acknowledge that these absolute numbers are influenced by changes in the population’s age distribution, as lung cancer incidence generally increases with age.

When examining gender, the age brackets recording the highest numbers of fatalities in 2019 were those aged 65–69 years and 70–74 years for both males and females. Males experienced 0.2375 million and 0.2322 million lung cancer deaths within the age groups of 65–69 and 70–74, respectively, contributing to 17.13% and 16.75% of all male deaths across all age groups. In parallel, for females in the same age categories, the death count stood at 0.1023 million and 0.0953 million, respectively, comprising 15.58% and 14.51% of total female deaths across all age groups (refer to Figure 1). For a more comprehensive analysis, we have adjusted the lung cancer mortality data for age-standardized mortality rates (ASMRs), which factor in the changing age distribution in the population. The ASMRs were used to compare mortality rates across different time points while accounting for changes in the population’s age structure.

#### 3.1.2. Global Age-Standardized Mortality Rates (ASMRs) for Lung Cancer from 1990 to 2019

The age-adjusted death rate refers to the mean individual mortality rate out of 100,000 individuals, factored by the proportion of individuals from respective age groups in the World Health Organization’s benchmark population (for further details, visit https://www.who.int/data/gho/indicator-metadata-registry/imr-details/78, accessed on 30 October 2023). According to the statistical data in Table 1 and Figure 2, the ASMR from lung cancer globally showed a decreasing trend from 27.3 per 100,000 in 1990 to 25.18 per 100,000 in 2019, representing a decrease of 7.77% over the period. From 1990 to 2019, the ASMR from lung cancer was consistently higher in males than females. However, there was a decreasing trend in the ASMR for males, while females showed an increasing trend (Table 1).

#### 3.1.3. Global and Regional AAPCs of Death from Lung Cancer from 1990 to 2019

The findings from the worldwide lung cancer Joinpoint regression model indicate that the standardized lung cancer incidence from 1990 to 2019 displays five turning points, or junction points, specifically occurring in 1990–1994 = 0.4 (95%CI = 0.1−0.7, *p* = 0.009); APC1994–1998 = −0.6 (95%CI = −1.0− −0.2, *p* = 0.010); APC1998–2002 = 0.1 (95%CI =−0.3–0.6, *p* = 0.482); APC2002–2010 = −0.3 (95%CI = −0.4− −0.1, *p* = 0.002); APC2010–2017 = −0.9 (95%CI = −1.1− −0.7, *p* < 0.001); and APC2017–2019 = 0.2, (95%CI = −1.5− −2.0, *p* = 0.771). Overall, the global AAPC within the period 1990–2019 was −0.3 (95%CI = −0.4− −0.1, *p* < 0.001), which means the normalized mortality rate of lung cancer declined by 0.3% per year on average between 1990 and 2019 (Figure 3).

The findings from the Joinpoint regression model across the 21 regions divided by GBD reveal that lung cancer mortality rates in regions such as Oceania, Southeast Asia, Southern and Eastern Asia, and Western and Eastern Sub-Saharan Africa were increasing. On the other hand, regions including Australasia, Central Asia, Central Latin America, Eastern Europe, High-income North America, and Southern Latin America displayed a declining trend, with the shift in trend being statistically significant (*p* < 0.01). For female standardized lung cancer death rates from 1990 to 2019, regions like Central Asia, Central Latin America, Eastern Europe, High-income Asia Pacific, and High-income North America showed a downturn, whereas the other regions exhibited an upward trend. The standardized mortality rate from lung cancer among men was decreasing in all regions apart from East Asia, Oceania, South Asia, Southeast Asia, and Western Sub-Saharan Africa, where the trend is on the rise (refer to Figure 4 and Table 2).

### 3.2. Average Annual Percentage Change (AAPC) in Lung Cancer Mortality Separated by Gender across Various Socio-Demographic Index (SDI) Regions

The outcomes of the Joinpoint regression model for regions with high SDI revealed a decline in the standardized lung cancer mortality rate from 1990 to 2019. This represented an average annual reduction of 0.81% (*t* = −3.48, *p* = 0). There was no statistical significance (APC = −0.1, *t* = −1.26, *p* = 0.222 and APC = 0.3, *t* = 0.38, *p* = 0.711); the average annual rates were 0.85% and 1.45% in 1995–2009 and 2009–2017, respectively. The percent change decreased slowly (*t* = −32.89, *p* = 0 and *t* = −15.827, *p* = 0). Between 1990 and 2019, the annual decline in the standardized mortality rate for lung cancer in males was by 1.52% while, in females, there was an annual rise of 0.3% (refer to Figure 5A and Table 2).

Findings from the Joinpoint regression analysis in high–middle-SDI regions indicated a decline in the age-adjusted death rate from lung cancer between 1990 and 2019, with an average yearly reduction of 0.43% (*t* = −5.8, *p* = 0). Among these values, the trend from 1998 to 2004 was not statistically significant (APC = 0.21, *t* = 1.69, *p* = 0.107), 1990–1994 exhibited an average yearly rate alteration of 1.28% (*t* = 4.58, *p* = 0), 1994–1998 and 2004–2019 exhibited average yearly rate alterations of 1.63%, and the average annual percentage change of 0.84% decreased slowly (*t* = −4.76, *p* = 0 and *t* = −17.47, *p* = 0). Over the span from 1990 to 2019, the age-adjusted death rate for lung cancer displayed a yearly reduction of 0.81% in males while demonstrating an annual increment of 0.67% in females (refer to Figure 5B and Table 2).

The findings from the Joinpoint regression analysis in the mid-level SDI areas showed a general reduction in the normalized mortality rate from lung cancer, with an average yearly increase of 0.65% from 1990 to 2019 (*t*-value = 11.89, *p*-value = 0). The trend was not statistically significant from 2004 to 2007 (APC = 0.25, *t* = 0.8, *p* = 0.44), declined slowly at an average annual percentage change of 0.296% from 2010 to 2019 (*t* = −7.27, *p* = 0), and increased slowly at an AAPC rate in the remaining years. The average annual increases between 1990 and 2019 were 0.65% for men and 0.81% for women (see Figure 5C and Table 2).

An increasing trend in the age-adjusted death rate in low-middle-SDI regions from 1990 to 2019 was observed after the Joinpoint regression analysis, with an average annual rise of 0.37% (*t* = 7.16, *p* = 0). The change trend was not statistically significant (APC = −0.28, *t* = −0.77, *p* = 0.45 and APC = 0.12, *t* = 2.12, *p* = 0.05); the increase rate was 0.39% in 1990–2001, and in 2004–2008 and 2015–2019, 0.81% and 0.77% of the average annual percentage change increased slowly (*t* = 12.03, *p* = 0, *t* = 4.79, *p* = 0 and *t* = 5.95, *p* = 0). From the period from 1990 to 2019, there were 0.20% and 1.16% annual increases in the age-adjusted death rates of lung cancer for males and females, respectively (refer to Figure 5D and Table 2).

The findings of the Joinpoint regression analysis in areas with a low Socio-Demographic Index (SDI) indicated that the age-adjusted death rate of lung cancer displayed an increasing pattern from 1990 to 2019, with an average yearly increment of 0.13% (*t* = 4.04, *p* = 0). There was no statistical significance (APC = −0.08, *t* = −1.27, *p* = 0.221 and APC = 0, *t* = 0.04, *p* = 0.965), and 0.2% and the average annual percentage change of 0.76% increased slowly (*t* = 3.33, *p* = 0.004, *t* = 3.9, *p* = 0.001 and *t* = 2.82, *p* = 0.012). Over the period 1990–2019, the AAPC in the normalized lung cancer mortality rate revealed a decline of 0.06% for males, while females saw an increase of 1.27% annually (refer to Figure 5E and Table 2).

### 3.3. Prediction of Global Lung Cancer Mortality from 2020 to 2030

Based on the predicted data, the ASMR of pulmonary cancer for males will reach up to 33.48/100,000 globally in 2030, showing a downward trend compared with 37.38/10 in 2019. The global ASMR for females is 15.83/100,000, which is on the rise compared with 14.99/100,000 in 2019 (Figure 6A–C). By 2030, it is projected that 2,126,900 people will die globally, of which 680,500 will be women; 1,446,400 will be men.

## 4. Discussion

Based on the data gleaned from this investigation, it is evident that global lung cancer fatalities in 2019 amounted to 2,042,600 cases, reflecting a notable increase of 91.75% when juxtaposed with the corresponding 1990 figure of 1,065,100 cases. The global ASDR attributed to lung cancer in 2019 stood at 25.18 per 100,000 individuals, with a noteworthy gender discrepancy of 37.38 per 100,000 for men and 14.99 per 100,000 for women. This gender-related contrast is indicative of a prevailing male preponderance; however, it is notable that the temporal trajectory reveals a decline in male mortality rates and an upswing in female mortality rates.

Regionally, the dynamics are multifaceted, with high-SDI and high–middle-SDI regions witnessing a diminishing trend in lung cancer mortality, marked by AAPCs of −0.3 and −0.8, respectively. In contrast, middle-SDI, low-SDI, and low-middle-SDI regions exhibited an upward trajectory in mortality rates, characterized by AAPC values of 0.1, 0.37, and 0.13, respectively, across the 21 regions classified by the GBD.

It is noteworthy that geographical disparities emerge, with discernible escalations in lung cancer mortality rates being observed in regions including Oceania, South Asia, East Asia, Western Sub-Saharan Africa, Southeast Asia, and Eastern Sub-Saharan Africa. Conversely, a declining pattern prevails in other regions. It is imperative to underscore the statistical validation (*p* < 0.05) of these trends.

While our study yields invaluable insights into the evolving trends and associated risk factors in lung cancer mortality, it is essential to exercise prudence when attributing direct causality between our findings and the effectiveness of specific measures. Indeed, the effectiveness of such measures remains a complex interplay deserving of further investigation and scrutiny.

### 4.1. Global Lung Cancer Deaths Are on the Rise

Lung cancer ranks among the most prevalent types of cancer globally, and it is associated with one of the highest death rates among malignant tumors. The mortality from lung cancer exhibits substantial regional disparities worldwide. The main causes of lung cancer are environmental factors such as smoking and environmental pollution. Especially in countries with rapid industrialization, environmental pollution is more likely to occur, which leads to an increase in lung cancer mortality.

Smoking stands out as a prominent risk factor for lung cancer. It plays a significant role in the elevated rates of lung cancer mortality observed worldwide. Smoking accounts for 80% of the causes of lung cancer [18]. It is known that the number of smokers aged 15 and over in the world is as high as 1.1 billion. Tobacco causes 8 million deaths worldwide every year. Approximately 1.2 million deaths have been attributed to exposure to secondhand smoke [19,20,21,22], of which more than 1 million have died from exposure to secondhand smoke [23,24]. In addition to smoking, environmental pollution has an impact on lung cancer mortality in global regions. Previous studies have demonstrated that the occurrence is associated with PM2.5 [25]; PM2.5 can be inhaled and deposited in the alveoli, which leads to allergic and inflammatory responses leading to irreversible changes in lung structure and function [26]. Ambient PM2.5-attributable lung cancer deaths and disability-adjusted mortality more than doubled from 1990 to 2019 [27]. The differences in environmental pollution levels and occupational exposures among different regions may explain the variation in the lung cancer trend in different regions, which may be caused by the different pollution levels and occupational exposures [28]. Additionally, existing patients may inherit the oncogene to their offspring, in which case their children will have high susceptibility to lung cancer [29]. According to the income level of a region, it also affects the mortality rate of lung cancer.

Most Economically Developed Countries (MEDCs) generally have more developed sanitation facilities, higher medical standards, and better health education. These factors contribute to the early diagnosis and treatment of lung cancer, thereby reducing mortality. Middle-income regions (Central Asia, Central Latin America, Eastern Europe) have relatively low sanitation facilities and medical resources, and there may be some unhealthy lifestyle habits and environmental factors that lead to an increase in lung cancer mortality. Moreover, low-income regions generally have problems such as high smoking rates, serious environmental pollution, insufficient sanitation facilities, and low levels of health education, leading to an increase in lung cancer mortality.

In addition, the global incidence of lung cancer increases with age, with the highest rates being in the 65–74 age group, which is associated with the fact that lung cancer takes decades to appear after the initiation of smoking and is therefore rare until the age of 30 years, peaking in the elderly [30], suggesting that population aging greatly increases mortality from lung cancer [31]. This may be related to the fact that as age increases, the body’s ability to recover from injury gradually decreases, making the burden of death increase as well.

To control the epidemiological trend, the implementation of preventive and controlled measures is essential. First, smoking control measures should be strengthened, including the implementation of anti-smoking policies, health education, and the provision of smoking cessation support services. At the same time, environmental protection efforts should be strengthened to reduce air pollution and the emissions of harmful substances. In addition, occupational health and safety management needs to be strengthened to reduce the risk of occupational exposure. Genetic counseling and early screening are also important measures that can help in the early detection and management of individuals with genetic susceptibility or familial lung cancer. Many countries are working on lung cancer screening programs aimed at the early detection of lung cancer and improving treatment success and the survival of patients. Global advances in lung cancer treatment and prevention require global cooperation and concerted efforts to alleviate the tremendous burden lung cancer places on society and families. Early diagnosis and treatment can improve the cure rate and reduce the mortality rate of pulmonary cancer in addition to early detection [31,32,33]. In addition, with the accelerated aging of the global population, older people with histories of smoking should be considered a priority population for lung cancer intervention [34]. What is more, tertiary prevention should be put into practice, which can improve the prognosis of patients and reduce their mortality rate.

Public health policies play a pivotal role in the fight against lung cancer. The burden of this disease is closely intertwined with modifiable risk factors, with smoking being a prominent one. As mentioned earlier, smoking is responsible for 80% of lung cancer cases worldwide. Therefore, it is imperative to address this risk factor through comprehensive smoking control measures.

In this context, governments and health authorities should implement and strengthen anti-smoking policies [35]. These policies should encompass not only awareness campaigns but also stringent regulations on tobacco sales, advertising, and public smoking areas. Health education programs targeting all age groups, particularly adolescents, are essential. These efforts should emphasize the hazards of smoking and provide information on smoking cessation support services.

Promoting smoking cessation is of utmost importance in preventing lung cancer [36]. The ‘smoking cessation’ approach should be central to any public health strategy aimed at reducing the incidence of this deadly disease. Effective smoking cessation programs, including counseling and access to nicotine replacement therapies, should be readily available to individuals seeking to quit smoking.

Early diagnosis is a key element in breaking the vicious circle of lung cancer mortality [37]. Lung cancer often presents with advanced stages, making it crucial to implement screening programs to detect the disease at an earlier, more treatable phase. These programs are vital, especially in regions with a high prevalence of smoking and environmental pollution. Lung cancer screening should be a priority in public health agendas. It involves the use of low-dose computed tomography (LDCT) scans for high-risk populations such as long-term smokers and individuals with a family history of lung cancer. The implementation of such programs has the potential to significantly reduce lung cancer-related deaths.

It is important to stress the significance of early intervention through screening. Patients diagnosed at an earlier stage have a higher chance of successful treatment and survival [38]. Hence, promoting and ensuring access to lung cancer screening for those at risk is a public health imperative.

While it has been well established that smoking plays a significant role in elevated lung cancer mortality, the effectiveness of measures to control this risk factor requires ongoing research and policy evaluation. Our study does not directly establish the causal impact of specific measures in reducing lung cancer mortality, although it emphasizes the importance of strengthening measures such as anti-smoking policies, health education, and environmental protection.

### 4.2. Lung Cancer Mortality in Women Is on the Rise and More Pronounced in Developed Countries

In 28 industrialized nations with high HDI (Human Development Index) ratings, including the United States, Canada, Australia, China, the majority of Western European countries, and Scandinavia, lung cancer has emerged as the leading cause of cancer-related fatalities among women. In 2018, Spanish researchers projected the ASMR of lung cancer by 2030 in 52 nations, comprising 33 high-income and 19 low-to-middle-income countries. The number of female lung cancer patients is predicted to be greater than the current number by 2030 [39].

However, the increasing mortality trend among females compared with males may be due to the increasing popularity of female smoking. In the United States in the 1920s and 1930s, targeted marketing focused on constructing society’s attitudes towards women’s smoking, deliberately raising levels of smoking by associating it with ideals of independence and liberation, including the notorious cigarette seen as “the torch of freedom”.

Meanwhile, peak smoking rates among men typically occurred in the 1950s and 1960s, while peak smoking rates among women occurred in the 1970s and 1980s. Therefore, the peak smoking rate of men occurred about 20 years earlier than that of women, but it should be noted that the specific time span of this gap may vary due to factors such as regions and cultural backgrounds [40,41,42]. The lack of women’s cognizance regarding lung cancer, coupled with a diminished focus on early screening and diagnosis, has resulted in diminished opportunities for the early detection and treatment of the disease in women. The biological traits of lung cancer in women differ from those in men, with non-small cell lung cancer frequently occurring in women, but its prognosis is relatively favorable [7].

Thus, screening and diagnosing lung cancer is essential to reducing mortality, in addition to controlling smoking and pollution. Lung cancer is often asymptomatic in its early stages, so lung cancer screening is recommended for high-risk groups such as long-term smokers and those with family histories of lung cancer. Modern medical technologies such as computed tomography (CT) scanning also provide better means for the early diagnosis of lung cancer. Meanwhile, strengthening public health education and conducting lung cancer prevention and control campaigns are also necessary to reduce lung cancer mortality in women.

Our study recognizes these trends and the potential reasons behind them but does not establish a direct link between these trends and the effectiveness of preventive measures. It underscores the importance of early diagnosis, smoking control, and lung cancer screening for reducing mortality in women.

### 4.3. Situational Analysis of Mortality across Several SDI Regions

The average shift in global lung cancer illustrates that high- and medium–high-SDI regions show a declining trend, whereas a rising trend has been observed in medium-, low–medium-, and low-SDI areas. It is anticipated that by 2030, regions with low to medium SDIs will account for 75% of all global cancer fatalities [43]. This is mainly related to the rising trend of smoking in these areas, the high incidence of air pollution, and insufficient lung cancer screening measures. At the same time, these regions also differ from other regions in terms of access to healthcare and treatment options [43]. High-SDI regions, when contrasted with middle-, low–middle-, and low-SDI areas, typically possess more robust healthcare infrastructures due to larger investments in them, education on healthy lifestyles, and improved living standards, which are potential geographical factors of lung cancer. Lung cancer mortality may show a downward trend in high-SDI regions and high–middle-SDI regions, which is the result of multiple factors [28]. Moreover, age is another risk factor. Population aging is an ongoing global issue that many countries suffer from, drawing the attention to whether they are at risk of developing cancer and to planning and managing their future health status. In terms of early lung cancer detection and treatment, quality healthcare and medical resources are important to improve patient survival [44]; in high and high–middle regions, lung cancer patients, even with other diseases, can receive prolonged medical care, thus improving their survival rates. Conversely, due to the relative deficiencies in healthcare resources, public health promotion, and environmental awareness in middle-, low–middle-, and low-SDI regions, coupled with lower per capita income levels and education levels, lung cancer mortality rates in these regions show an increasing trend [28]. Due to the lack of investment in early cancer screening, treatment, and management in these regions, as well as the relatively low self-care awareness and health knowledge of patients, a large number of lung cancer patients are unable to receive timely and effective treatment [45]. In addition, in some low-SDI regions, people’s living environments are relatively harsh, including the widespread presence of flue-cured tobacco, haze, and other pollutants. Such elements could potentially escalate the incidence and mortality related to lung cancer in these regions [45]. As for lung cancer management, advancements in treatment methodologies, such as targeted therapies or immunotherapy, have been seen for both small cell [46] and non-small cell lung cancers [47]. These innovative treatments contribute to the progressive enhancement of patient survival rates. However, in low- and middle-income countries, many cancer patients, including lung cancer patients, cannot obtain these standardized treatment options [48] or even traditional radiotherapy, which is not fully provided in low–middle regions due to equipment problems [49]. Therefore, the global average change trend of lung cancer mortality is not only related to the incidence factors and screening measures in different regions but is also related to different treatment options.

### 4.4. Global Projections of Lung Cancer Mortality

There has been a slow upward trend in female mortality and a significant downward trend in male mortality from 2020 to 2030. This indicates the initial success of the Framework Convention on Tobacco Control [28], the first global public health treaty. However, the gradual slow increase in mortality among women may be related to factors such as social changes, advertising, and marketing strategies, in addition to changes in smoking habits [40,41,42]. It may also be related to screening and diagnostic disparities, with a relatively low prevalence of lung cancer screening compared to other cancers such as breast cancer. Universal screening and attention for breast cancer may lead to earlier diagnosis and treatment for women, which may improve survival. However, lung cancer screening is relatively under-promoted and may lead to later detection of lung cancer in women, thereby affecting treatment outcomes and survival. Therefore, in order to reduce the mortality rate of female lung cancer, more efforts should be made to educate and publicize the preventive measures and screening methods related to the prevention and treatment of lung cancer.

## 5. Conclusions

This study, spanning nearly three decades, encompassed the comprehensive collection and analysis of global and regional lung cancer mortality data extracted from the GBD 2019 database. The investigation scrutinized temporal trends in lung cancer mortality at both the global and regional levels, revealing intriguing variations in age-standardized lung cancer mortality rates. The overarching global ASMR for lung cancer displayed a relatively stable trajectory. However, discernible disparities emerged when scrutinizing these trends at a regional scale, underscoring the critical significance of discerning and addressing distinct regional patterns and idiosyncratic challenges.

An unmistakable gender disparity in lung cancer mortality rates became evident. Notably, a diminishing trend in age-standardized mortality among males stood in contrast to a discernible upswing among females. The underpinning rationale for this gender contrast is complex and multifaceted, potentially attributed to differential smoking behaviors, distinct environmental exposures, and gender-specific biological factors. These findings accentuate the vital importance of adopting precisely tailored strategies that effectively address the unique risk factors and challenges confronting each gender.

Lung cancer mortality exhibited a notable correlation with the socioeconomic development of regions. Areas characterized by lower socioeconomic development levels witnessed markedly elevated lung cancer mortality rates. This phenomenon can be attributed to higher prevalences of smoking, underdeveloped healthcare infrastructures, and a lower level of public awareness concerning the risks associated with lung cancer. In stark contrast, regions classified as high-SDI areas demonstrated substantially lower mortality rates, primarily attributed to the successful implementation of stringent tobacco control measures, improved healthcare infrastructures, and elevated health literacy. Consequently, the imperative to combat lung cancer mortality lies in the stringent enforcement of tobacco control policies, the promotion of efficacious smoking cessation initiatives, and the enhancement of public awareness campaigns outlining the perils of smoking.

Elevating the standard of healthcare, instituting a robust screening program, and ensuring the prompt diagnosis and treatment of lung cancer emerge as pivotal steps in the collective endeavor to mitigate lung cancer mortality. In summation, this study underscores the profound interdependence of public health policies, smoking cessation efforts, and the imperative role of lung cancer screening programs in our combat against this formidable disease. The active promotion and seamless integration of these measures into healthcare systems on a global scale is indispensable. A unified collaboration between governments, healthcare institutions, and the international community is of paramount significance in alleviating the staggering burden posed by this devastating ailment.

Nevertheless, several limitations have come to our attention in the course of this analysis. Our approach primarily focused on statistical trends without offering an exhaustive exploration of underlying risk factors contributing to the landscape of global lung cancer mortality. Future investigations should encompass a comprehensive evaluation of region-specific risk factors, including but not limited to smoking behaviors, environmental exposures, and genetic predispositions. Secondly, the temporal scope of our study was bound by the availability of data, leading to the omission of the most recent lung cancer mortality statistics. Subsequent research endeavors should prioritize the utilization of up-to-date data to ensure a more precise depiction of evolving global trends in lung cancer mortality. Acknowledging these constraints, it is imperative that forthcoming studies delve deeper into the complexities of lung cancer risk factors and gender disparities. Furthermore, the incorporation of the most current data is paramount for a more profound comprehension of global lung cancer mortality trends, enabling the enhanced monitoring and assessment of the effectiveness of preventive and control measures.

## Figures and Tables

**Figure 1 healthcare-11-02920-f001:**
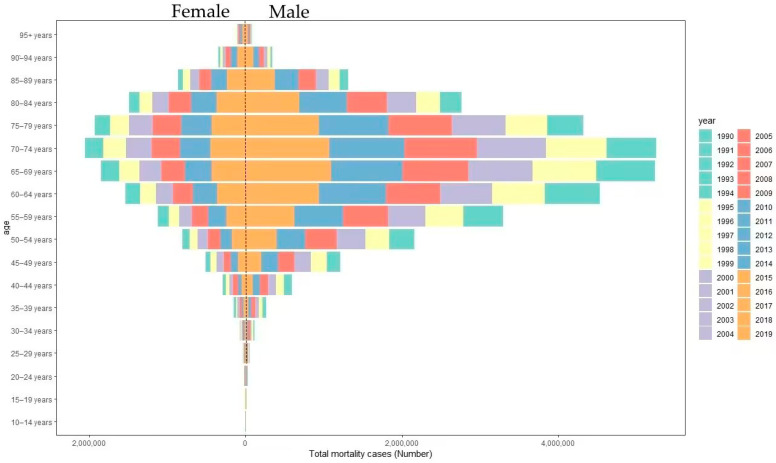
Lung cancer mortality cases by age group from 1990 to 2019.

**Figure 2 healthcare-11-02920-f002:**
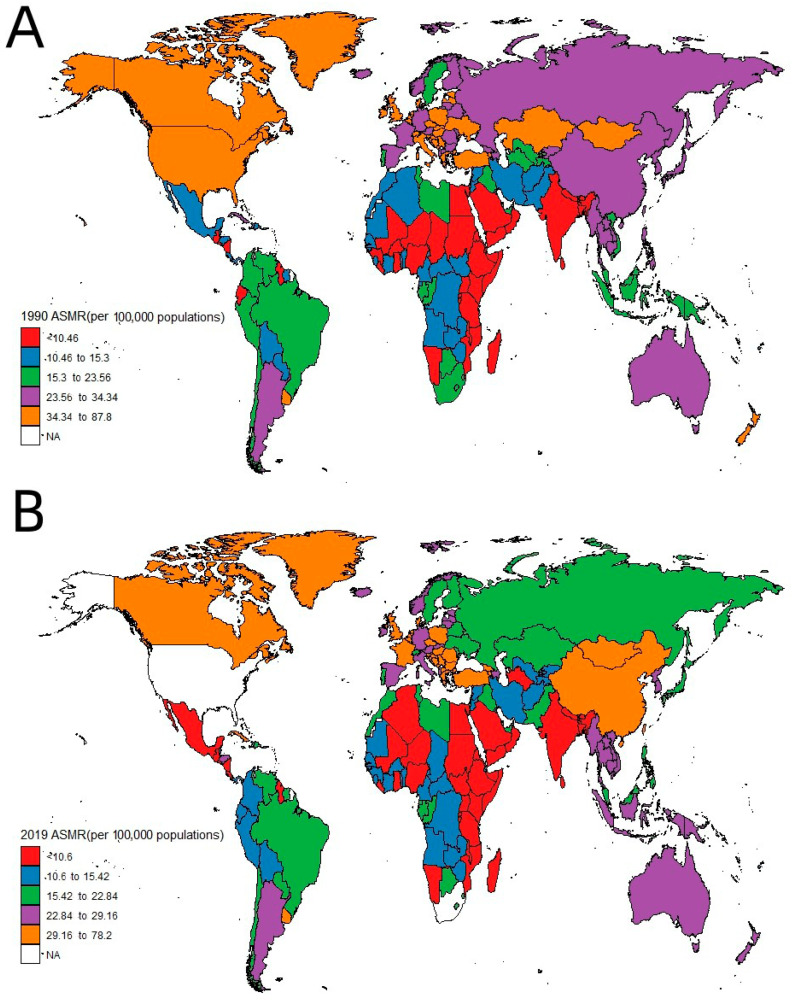
Map of ASMRs of lung cancer per 100,000 individuals by country: (**A**) 1990, (**B**) 2019. (NA refers to Not Available).

**Figure 3 healthcare-11-02920-f003:**
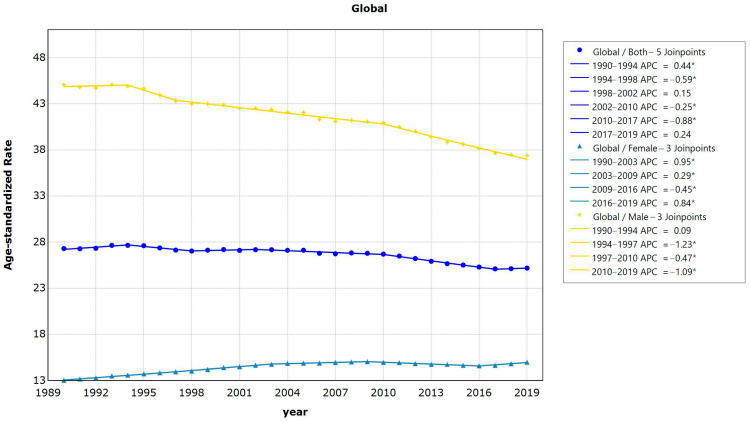
Results of Joinpoint regression analysis of trends in global lung cancer mortality data, 1990−2019. (* Indicates that the Annual Percent Change (APC) is significantly different from zero at the alpha = 0.05 level).

**Figure 4 healthcare-11-02920-f004:**
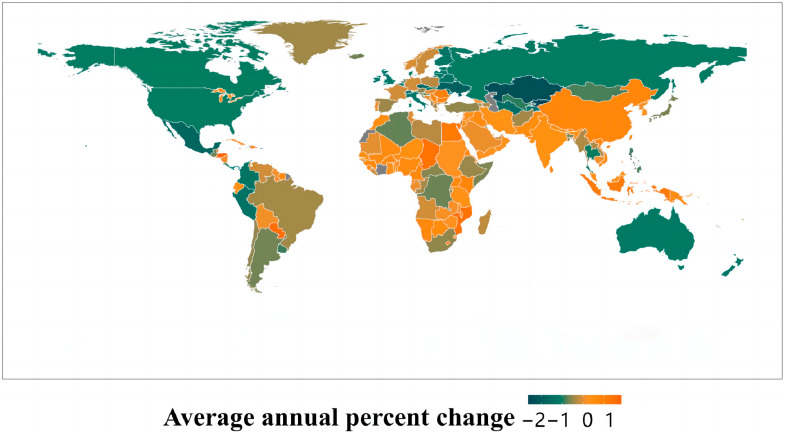
Map of the global and region AAPCs of lung cancer ASMRs from 1990 to 2019 in males and females.

**Figure 5 healthcare-11-02920-f005:**
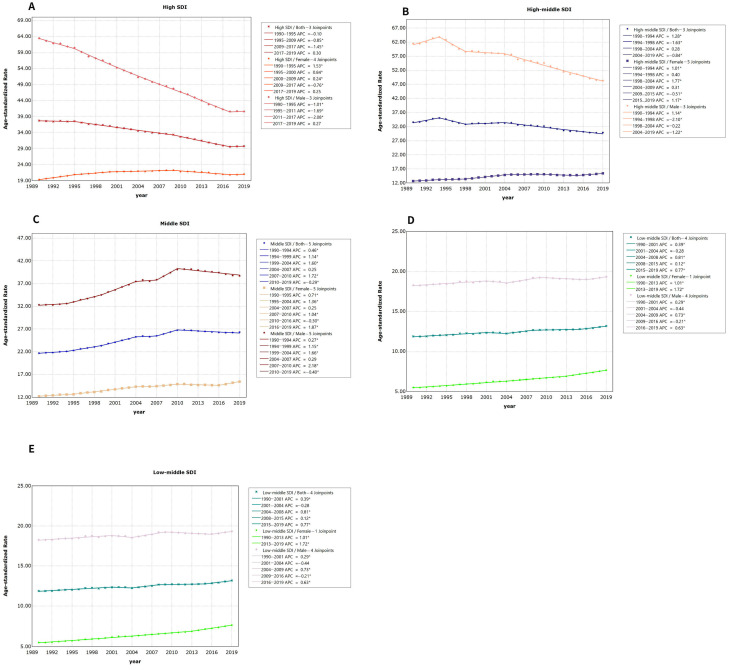
Results of Joinpoint regression analysis of trends in different SDI regions’ lung cancer mortality data, 1990−2019: (**A**): high SDI; (**B**): high–middle SDI; (**C**): middle SDI; (**D**): low–middle SDI; (**E**): low SDI. (* Indicates that the Annual Percent Change (APC) is significantly different from zero at the alpha = 0.05 level).

**Figure 6 healthcare-11-02920-f006:**
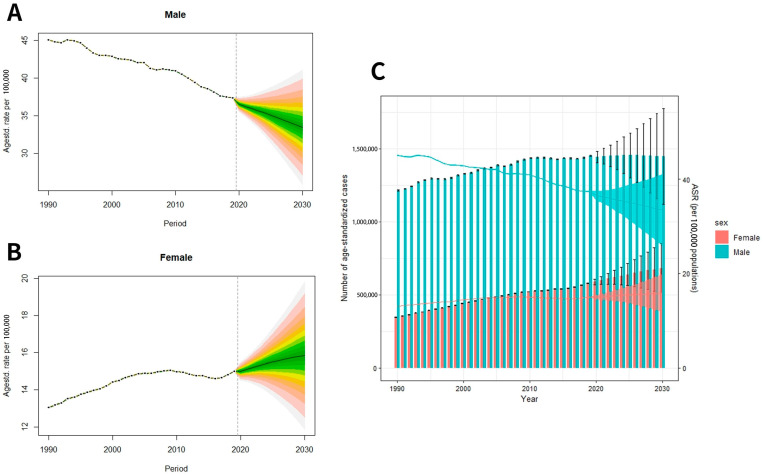
Trends in lung cancer mortality rate by sex: (**A**) male, (**B**) female; (**C**) trends of global lung cancer mortality numbers and age-standardized mortality rates. (The meaning of the different gradient colors in (**A**,**B**): The outermost ring represents the 95% confidence interval, and then each color gradient gradients inward at a rate of 0.05%).

**Table 1 healthcare-11-02920-t001:** The number and age-standardized rate of lung cancer deaths, globally, from 1990 to 2019.

Characteristics/Year	Deaths					
	Male		Female		Both	
	Number (95%UI)	Age-Standardized Rate per 100,000, No. (95%UI)	Number (95%UI)	Age-Standardized Rate per 100,000, No. (95%UI)	Number (95%UI)	Age-Standardized Rate per 100,000, No. (95%UI)
1990	790,748 (749,888–838,458)	45.08 (42.78–47.59)	274,391 (257,392–289,938)	13.04 (12.2–13.76)	1,065,139 (1,019,217–1,117,181)	27.3 (26.03–28.59)
1994	865,102 (830,724–900,778)	44.93 (43.04–46.73)	311,655 (294,382–327,630)	13.6 (12.82–14.29)	1,176,758 (1,134,265–1,216,665)	27.65 (26.6–28.6)
1999	923,809 (891,555–958,118)	43 (41.33–44.59)	364,000 (343,803–383,041)	14.2 (13.36–14.93)	1,287,809 (1,244,696–1,328,637)	27.12 (26.11–28.01)
2004	1,020,521 (977,312–1,064,795)	42.07 (40.15–43.81)	428,620 (402,212–452,741)	14.85 (13.91–15.68)	1,449,142 (1,388,504–1,503,886)	27.11 (25.88–28.14)
2009	1,136,532 (1,082,355–1,185,206)	41.06 (39.04–42.93)	495,246 (461,780–522,617)	15.04 (14.01–15.87)	1,631,778 (1,552,380–1,693,514)	26.79 (25.34–27.84)
2014	1,240,019 (1,169,680–1,313,650)	38.83 (36.59–41.12)	559,284 (516,144–593,643)	14.75 (13.61–15.65)	1,799,303 (1,700,096–1,890,259)	25.66 (24.15–26.98)
2019	1,386,094 (1,260,237–1,513,800)	37.38 (34.09–40.74)	656,546 (590,247–718,975)	14.99 (13.48–16.41)	2,042,640 (1,879,241–2,193,269)	25.18 (23.16–27.01)

**Table 2 healthcare-11-02920-t002:** Changes in the global and regional AAPCs of lung cancer from 1990 to 2019.

Characteristics/Sex	AAPCs		
	Both (95%CI)	Female (95%CI)	Male (95%CI)
**Global**	−0.27 (−0.32 to −0.24)	0.46 (0.43 to 0.49)	−0.66 (−0.70 to −0.63)
**Socio-Demographic Index**			
High SDI	−0.81 (−0.86 to −0.78)	0.28 (0.24 to 0.31)	−1.52 (−1.57 to −1.48)
High–middle SDI	−0.43 (−0.49 to −0.36)	0.67 (0.63 to 0.70)	−0.81 (−0.88 to −0.75)
Middle SDI	0.65 (0.62 to 0.67)	0.81 (0.77 to 0.84)	0.65 (0.62 to 0.68)
Low–middle SDI	0.37 (0.34 to 0.40)	1.16 (1.12 to 1.20)	0.20 (0.18 to 0.23)
Low SDI	0.15 (0.13 to 0.17)	1.27 (1.25 to 1.30)	−0.06 (−0.08 to −0.05)
**Region**			
Andean Latin America	−0.71 (−0.83 to −0.61)	0.22 (−0.02 to 0.38)	−1.24 (−1.35 to −1.15)
Australasia	−1.14 (−1.20 to −1.10)	0.23 (0.17 to 0.28)	−2.02 (−2.12 to −1.96)
Caribbean	−0.04 (−0.11 to 0.02)	0.59 (0.50 to 0.66)	−0.26 (−0.36 to −0.16)
Central Asia	−1.45 (−1.48 to −1.41)	−0.93 (−1.00 to −0.86)	−1.58 (−1.62 to −1.54)
Central Europe	−0.09 (−0.14 to −0.05)	1.67 (1.48 to 1.79)	−0.64 (−0.70 to −0.58)
Central Latin America	−1.11 (−1.29 to −1.02)	−0.43 (−0.48 to −0.38)	−1.27 (−1.48 to −1.14)
Central Sub-Saharan Africa	−0.51 (−0.54 to −0.47)	0.65 (0.63 to 0.69)	−0.58 (−0.61 to −0.55)
East Asia	0.77 (0.70 to 0.83)	0.72 (0.65 to 0.78)	0.79 (0.74 to 0.83)
Eastern Europe	−1.17 (−2.71 to −0.72)	−0.77 (−0.92 to −0.62)	−1.44 (−1.87 to −0.93)
Eastern Sub-Saharan Africa	0.12 (0.09 to 0.15)	1.22 (1.20 to 1.25)	−0.07 (−0.08 to −0.05)
High-income Asia Pacific	−0.54 (−0.59 to −0.51)	−0.29 (−0.33 to −0.26)	−0.78 (−0.85 to −0.73)
High-income North America	−1.07 (−1.13 to −1.03)	−0.18 (−0.23 to −0.15)	−1.77 (−1.84 to −1.71)
North Africa and Middle East	−0.01 (−0.05 to 0.05)	1.44 (1.38 to 1.50)	−0.34 (−0.39 to −0.28)
Oceania	0.36 (0.35 to 0.38)	0.80 (0.78 to 0.82)	0.24 (0.22 to 0.26)
South Asia	0.24 (0.18 to 0.32)	1.73 (1.58 to 1.90)	0.03 (−0.06 to 0.12)
Southeast Asia	0.22 (0.19 to 0.25)	0.75 (0.73 to 0.77)	0.06 (0.03 to 0.09)
Southern Latin America	−0.89 (−0.95 to −0.84)	1.22 (1.18 to 1.25)	−1.43 (−1.53 to −1.35)
Southern Sub-Saharan Africa	−0.30 (−0.50 to −0.08)	0.54 (0.37 to 0.74)	−0.44 (−0.66 to −0.22)
Tropical Latin America	−0.37 (−0.42 to −0.33)	0.83 (0.76 to 0.89)	−0.96 (−1.01 to −0.91)
Western Europe	−0.69 (−0.74 to −0.67)	1.19 (1.17 to 1.21)	−1.53 (−1.59 to −1.50)
Western Sub-Saharan Africa	0.62 (0.60 to 0.63)	1.26 (1.24 to 1.29)	0.46 (0.44 to 0.47)

## Data Availability

The data come from the 2019 GBD database, which is the world’s most comprehensive catalog of surveys, censuses, vital statistics, and other health-related data.

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
