# Peer review of "Epidemiological Analysis of Global and Regional Lung Cancer Mortality: Based on 30-Year Data Analysis of Global Burden Disease Database"

_healthcare, 2023, doi:10.3390/healthcare11222920_

Round 1

Reviewer 1 Report

Comments and Suggestions for Authors

Thank you very much for the opportunity to review your manuscript. The objective of this study was to understand dynamic global and regional lung cancer fatality trends and provide a foundation for effective global lung cancer prevention and treatment strategies. It is recommended that the author supplement the research background, such as comparing the use of relevant research methods, in order to highlight the advantages of the methods used in this study.

Reviewer 2 Report

Comments and Suggestions for Authors

Strengths: This epidemiological analysis addresses the evolution of Lung Cancer cases in different regions and gives a global perspective of the issue. The methodology used was appropriate and identified the gap between developed and developing countries. Furthermore, the authors offered suggestions on how to improve the situation. A very important question is how will developing countries make a difference with the lack of resources.  

Weaknesses: There’s a clear link with public health policies. The ‘smoking cessation’ approach is vital to prevent. Lung Cancer screening programs are needed for early diagnosis. The authors need to stress the importance of screening and early intervention to break this vicious circle.

Figure 2: Please format image, distorted.

Figure 4: Please format image, distorted

Is there a copyright on these images?

Figure 5: Difficult to read, might need to increase size of charts

Overall, a very comprehensive review of current world epidemiology of lung cancer.

Reviewer 3 Report

Comments and Suggestions for Authors

Thanks for the paper on an important cancer of massive global burden. This paper used Global Burden of Disease (GBD) 2019 database to study the trend between 1990 and 2019. The Age-Standardized Mortality Rates (ASMR) was analysed by trend, geographic region, socio-demographic index (SDI) and gender. Projection of the ASMR was also included in this paper. 

In addition to typos, there are a few things for the authors to clarify, as follows:

Figure 1 is not "Lung cancer mortality by age group in 2019", as this Figures contains multiple years data, so please either change the title of the Figure or change the Figure.

Figure 2 does not have info by gender, so it does not support "there was a decreasing trend in the ASMR for males, while females showed an increasing trend". I wonder if you could include the maps by gender. 

Figure 6 C can be simplified as it does not need to include the info already included in Figure 6 A and Figure 6 B.

3.1.3 is mainly about the data by geographic region and 3.2 for SDI region, please make it clear to the readers. 

Lines 254-259 does not make sense to me, so it should be better reworded. 

Some minor typos, as follows:

Please specify 'AAPC' in the Abstract.

Line 37, please remove "which accounted for".

Line 70, please replace . (full stop) with , (comma).

Line 74, please provide the correct webpage.

Line 128, please include . (full stop) before "According to". 

Please indicate Figure 4 is both Males and Females.

Line 226: ASDM? or ASMR?

Lines 257-259, please include a reference.

Lines 290-295, this sentence is too long and needs to be reworded. 

Line 304, please replace 'higher' with 'increasing'.

Comments on the Quality of English Language

English language was ok but it would be better if it can be further improved so that it can be more natural, e.g. with the support of a native speaker. 

Reviewer 4 Report

Comments and Suggestions for Authors

Major comments

1.       Lung cancer mortality

How did the authors consider the effects of increased life expectancy and changes in the population's age distribution in comparing lung cancer mortality cases and rates? This is relevant not only to this part of the study but also to the overall analysis of the study.

2.       2.3. Bayesian age-period-cohort analysis

Please provide a supplemental description of this analysis to ensure reproducibility. Please also indicate the software and version used in the analysis.

3.       Discussion

Please make a clear distinction between the proven findings and the author's opinion. The objective of this study, "effective measures," is not necessarily demonstrated by the analysis of this study.

Minor comments

1.       The title of Figure 1 should be "Lung cancer mortality cases by age group in 2019".

2.       The aspect ratio of the maps in Figures 2 and 4 appears distorted and should be corrected.

Round 2

Reviewer 4 Report

Comments and Suggestions for Authors

I think the quality of the paper has improved.